

# HaloTag is an effective expression and solubilisation fusion partner for a range of fibroblast growth factors

Changye Sun[1,3], Yong Li[1,3], Sarah E. Taylor[1], Xianqing Mao[2], Mark C. Wilkinson[1] and David G. Fernig[1]

[1] Department of Biochemistry, Institute of Integrative Biology, University of Liverpool, Liverpool, UK
[2] Department of Oncology, Laboratory of Cellular and Molecular Oncology, Luxembourg Institute of Health, Luxembourg
[3] These authors contributed equally to this work.

Corresponding author
David G. Fernig, dgfernig@liv.ac.uk

## ABSTRACT

The production of recombinant proteins such as the fibroblast growth factors (FGFs) is the key to establishing their function in cell communication. The production of recombinant FGFs in *E. coli* is limited, however, due to expression and solubility problems. HaloTag has been used as a fusion protein to introduce a genetically-encoded means for chemical conjugation of probes. We have expressed 11 FGF proteins with an N-terminal HaloTag, followed by a tobacco etch virus (TEV) protease cleavage site to allow release of the FGF protein. These were purified by heparin-affinity chromatography, and in some instances by further ion-exchange chromatography. It was found that HaloTag did not adversely affect the expression of FGF1 and FGF10, both of which expressed well as soluble proteins. The N-terminal HaloTag fusion was found to enhance the expression and yield of FGF2, FGF3 and FGF7. Moreover, whereas FGF6, FGF8, FGF16, FGF17, FGF20 and FGF22 were only expressed as insoluble proteins, their N-terminal HaloTag fusion counterparts (Halo-FGFs) were soluble, and could be successfully purified. However, cleavage of Halo-FGF6, -FGF8 and -FGF22 with TEV resulted in aggregation of the FGF protein. Measurement of phosphorylation of p42/44 mitogen-activated protein kinase and of cell growth demonstrated that the HaloTag fusion proteins were biologically active. Thus, HaloTag provides a means to enhance the expression of soluble recombinant proteins, in addition to providing a chemical genetics route for covalent tagging of proteins.

## INTRODUCTION

Of the 18 receptor-binding fibroblast growth factors (FGF), 15 also bind a heparan sulfate co-receptor and are classed as growth factors and morphogens. These are grouped into 5 subfamilies based on their protein sequence similarity (*Itoh, 2007*; *Ornitz, 2000*), and they regulate a myriad of processes in development, homeostasis and in some diseases (*Beenken & Mohammadi, 2009*; *Turner & Grose, 2010*). Recombinant FGFs provide a key tool to study their structure–function relationships, and labelling FGFs for microscopy

has been important in probing the mechanisms of, for example, their transport (*Duchesne et al., 2012*; *Lin, 2004*; *Yu et al., 2009*). Chemical labelling has disadvantages compared to genetically encoded labelling, since with the latter it is easier to predict the structural and hence functional consequences of labelling, which can be achieved both *in vitro* and *in vivo*. While fluorescent proteins remain a mainstay of genetic labelling, they have limitations. These have been overcome, for example, by non-covalent tagging of proteins on hexahistidine sequences with Tris-Ni$^{2+}$ nitriloacetic acid (*Huang et al., 2009*; *Lata et al., 2005*; *Tinazli et al., 2005*), which has allowed diverse labelling strategies, ranging from fluorescent dyes (*Uchinomiya et al., 2009*) and quantum dots (*Roullier et al., 2009*; *Susumu et al., 2010*) to gold nanoparticles (*Duchesne et al., 2008*). However, non-covalent coupling is reversible and exchange may occur in this instance with histidine-rich patches on endogenous proteins.

HaloTag is a mutant of a bacterial haloalkane dehalogenase, which reacts with chloroalkane ligands to form a covalent bond that represents the covalent intermediate of the enzyme's normal catalytic cycle (*Los et al., 2008*). Fluorescent dyes (*Los et al., 2008*) and quantum dots (*Zhang et al., 2006b*) carrying a chloroalkane group have been used to label HaloTag fusion proteins for fluorescence imaging. This approach is particularly versatile, since it combines the power of a genetically encoded tag (the HaloTag protein) with covalent labelling.

Consequently, we set out to produce N-terminal HaloTag fusions of different FGFs. In the course of this work, we observed that the N-terminal HaloTag fusion had a substantial effect on the expression of the more recalcitrant FGFs, consistent with the observation that HaloTag is a potential solubilisation tag for recombinant proteins (*Ohana et al., 2009*). Thus, whereas expression of FGF1 and FGF10 was somewhat reduced and that of FGF2 increased, expression of FGF7, which can be toxic (*Ron et al., 1993*), was no longer so, while expression of soluble FGF3, FGF6, FGF7, FGF8, FGF16, FGF17, FGF20 and FGF22 was markedly enhanced. This is in contrast to previous reports where FGFs such as FGF6 (*Pizette et al., 1991*), FGF8 (*Loo & Salmivirta, 2002*; *Macarthur et al., 1995*; *Vogel, Rodriguez & IzpisuaBelmonte, 1996*), FGF16 (*Danilenko et al., 1999*) and FGF20 (*Jeffers et al., 2002*; *Kalinina et al., 2009*) have been found to be mainly expressed in inclusion bodies, even as truncated proteins, and so require refolding. Thus, HaloTag provides not just a means to label proteins covalently and specifically, but is also a useful solublisation partner for the production of recombinant proteins.

## MATERIALS AND METHODS

### Materials

pET-14b vectors containing cDNAs encoding FGF1 and FGF2 and pET-M11 vector containing FGF7 cDNA were as described (*Xu et al., 2012*); cDNAs encoding FGF3, FGF10, FGF16, FGF17 and FGF20 were purchased from Eurofins Genomics (Ebersberg, Germany); cDNA encoding FGF6, FGF8 and FGF22 were purchased from Life Technologies (Paisley, UK); cDNAs encoding HaloTag was acquired from Kazusa DNA Research Institute (Kisarazu, Japan); Primers for PCR were from Life Technologies (Paisley, UK). All

**Table 1** **Peptide sequences of FGFs, the N-terminal HisTag constructs and the N-terminal HaloTag constructs.** FGF names, sequences and amino acid numbering are according to the UniProt entry. FGF1 is an N-terminal truncated protein (*Ke et al., 1990*). FGF2 does not possess a secretory signal sequence, whereas there is no signal peptide recognised in Uniprot for FGF16 and FGF20; consequently full length protein sequence was expressed. For all other FGFs, the protein expressed was without the Uniprot determined secretory signal sequence. FGFx refers any one of the FGFs. TEV cleavage sites are in red.

| Name | UniProt accession number | Residues in mature protein |
|---|---|---|
| FGF1 | P05230 | 16–155 |
| FGF2 | P09038-2 | 1–155 |
| FGF3 | P11487 | 18–239 |
| FGF6 | P10767 | 38–208 |
| FGF7 | P21781 | 32–194 |
| FGF8b | P55075-3 | 23–215 |
| FGF10 | O15520 | 38–208 |
| FGF16 | O43320 | 1–207 |
| FGF17 | O60258-1 | 23–216 |
| FGF20 | Q9NP95 | 1–211 |
| FGF22 | Q9HCT0 | 23–170 |
| HisTag terminus (pET-M11) | | MKHHHHHHPMSDYDIPTTENLYFQGA-[FGFx] |
| HaloTag and TEV site to conjoin with FGF sequence | | MPEIGTGFPFDPHYVEVLGERMHYVDVGPRDGTPVLFLHGNPTSSYV WRNIIPHVAPTHRCIAPDLIGMGKSDKPDLGYFFDDHVRFMDAFIEAL GLEEVVLVIHDWGSALGFHWAKRNPERVKGIAFMEFIRPIPTWDEWPE FARETFQAFRTTDVGRKLIIDQNVFIEGTLPMGVVRPLTEVEMDHYREP FLNPVDREPLWRFPNELPIAGEPANIVALVEEYMDWLHQSPVPKLLFWG TPGVLIPPAEAARLAKSLPNCKAVDIGPGLNLLQEDNPDLIGSEIARWLS TLEISGEPTTEDLYFQS-[FGFx] |

of the protein sequences corresponding to the above cDNAs are listed in Table 1. Enzymes for cloning were from: NcoI, BamHI and T4 ligase (NEB, Hitchin, UK); KOD Hot Start DNA polymerase (Merck, Hertfordshire, UK); In-Fusion® HD Cloning Kit (Clontech, Takara Bio Europe SAS, Saint-Germain-en-Laye, France). Bacterial cells: DH5$\alpha$, BL21 (DE3) pLysS and SoluBL21 were a gift from Olga Mayans, University of Liverpool. The sources of other materials were as follows: LB broth and LB agar (Merck, Hertfordshire, Germany); Soniprep 150 Plus (MSE, London, UK); Affi-Gel® Heparin Gel (Bio-Rad, Hertfordshire, UK), CM Sepharose Fast Flow, DEAE Sepharose Fast Flow, HiTrap Q HP column; empty disposable PD-10 Columns; ÄKTApurifier 100 plus (GE Healthcare, Buckinghamshire, UK).

For cell culture the following materials were used: Dulbecco's modified Eagle's medium (DMEM, Life Technologies), fetal calf serum (FCS, Labtech International Ltd, East Sussex, UK), 7.5% (w/v) sodium bicarbonate (Invitrogen, Paisley, UK), 200 mM L-glutamine (Gibco), 5 µg/mL insulin (Sigma-Aldrich, Dorset, UK), 5 µg/mL hydrocortisone (Sigma-Aldrich), bovine serum albumin (BSA; A7030, Sigma-Aldrich) for cell culture, cell culture dishes (Corning, Nottingham, UK). For SDS-PAGE and Western blotting: dried skimmed milk (Marvel, Spalding, UK), BSA (Fisher Scientific, Loughborough, UK), protease inhibitor (Roche, Burgess Hill, UK), phospho-p44/42 MAPK (T202/Y204) antibody (Cell Signalling, NEB, Hitchin, UK), monoclonal $\beta$-actin antibody

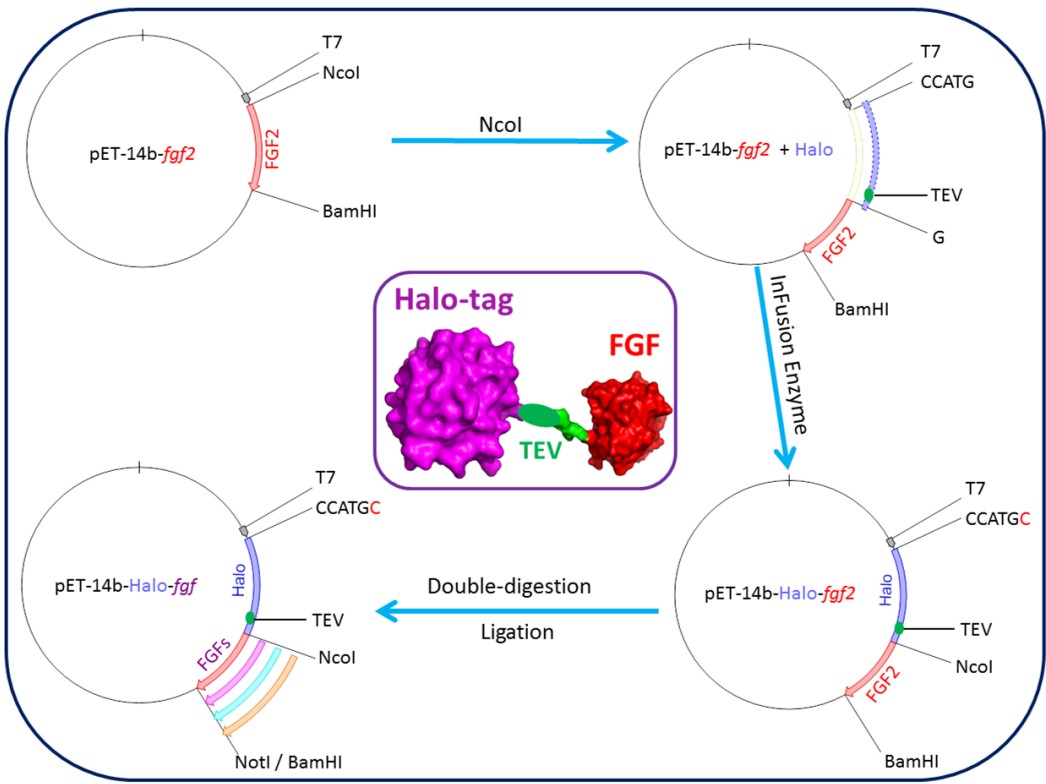

**Figure 1 Cloning strategy for plasmids encoding Halo-FGFs.** DNA encoding HaloTag was inserted 5′ of the FGF2 coding sequence with the In-Fusion HD enzyme. Subsequently, a NotI cleavage site was added 5′ to the BamHI site and other FGFs were exchanged into the plasmid using the digestion-ligation cloning method. A cartoon structure of Halo-FGF is presented in the middle of this figure.

(Sigma-Aldrich, Dorset, UK), anti-mouse IgG, horseradish peroxidase-linked antibody (Cell Signalling, NEB), polyvinylidene fluoride (PVDF) transfer membrane (Millipore UK, Hertfordshire, UK), enhanced chemiluminescence (ECL) Western blotting reagents (GE Healthcare, Little Chalfont, UK), Hyperfilm (GE Healthcare, Little Chalfont, UK).

## DNA cloning of hexahistidine tagged FGFs (His-FGFs) and HaloTag tagged FGFs (Halo-FGFs)

DNA encoding FGF1, FGF3, FGF6, FGF8, FGF10, FGF16, FGF17, FGF20 and FGF22 was cloned into pET-M11 such that the resulting protein would have a N-terminal 6xhis tag followed by a tobacco etch virus (TEV) cleavage site (ENLYFQ). FGF2 and FGF7 DNA sequences were previously cloned into pET-14b and pET-M11, respectively (*Xu et al., 2012*).

A plasmid encoding Halo-FGF2 was produced by adding a HaloTag encoding DNA sequence in-frame 5′ to a DNA sequence encoding full-length FGF2. This construct was then used to produce the other DNAs encoding Halo-FGFs (Fig. 1). The plasmid pET-14b-*fgf*2 contains NcoI and BamHI cleavage sites 5′ and 3′ of *fgf*2, respectively. This vector was linearized by digestion with NcoI. The DNA encoding HaloTag (Fig. 1: blue insert) was amplified by PCR using the Halo-FGF2-Forward, AAGGAGATATA CCATGCCAGAAATCGGTACTG, and Halo-FGF2-Reverse,

**Table 2 Concentrations of NaCl in 50 mM Tris-Cl buffer (pH 7.4) used for heparin affinity chromatography of FGFs.** [NaCl] for lysate is the concentration of NaCl in the sample applied to the column.

| Name | [NaCl] for lysate (M) | [NaCl] for wash (M) | [NaCl] for elution (M) |
|---|---|---|---|
| FGF1 | 0.6 | 0.6 | 1.0 |
| FGF2 | 0.6 | 0.6 | 1.5 |
| FGF3 | 0.3 | 0.6 | 1.0 |
| FGF6 | 0.3 | 0.4 | 1.0 |
| FGF7 | 0.3 | 0.3 | 1.0 |
| FGF8 | 0.6 | 0.6 | 1.5 |
| FGF10 | 0.6 | 0.6 | 1.0 |
| FGF16 | 0.3 | 0.4 | 1.0 |
| FGF17 | 0.6 | 0.6 | 1.0 |
| FGF20 | 0.3 | 0.4 | 1.0 |
| FGF22 | 0.6 | 0.8 | 1.5 |

TCCCGGCTGCCATGGAGCTCTGAAAGTACAGATC, primers (NcoI cleavage site underlined), and inserted into the linearized vector using In-Fusion enzyme. A TEV cleavage site (Fig. 1: green ellipsoid) was also included at the C-terminus of HaloTag to allow release of the FGF. A NotI cleavage site was also inserted 5′ of the BamHI to provide an additional 3′ cleavage sites for cloning. The other cDNAs (FGF1, FGF3, FGF6, FGF7, FGF8, FGF10, FGF16, FGF17, FGF20 and FGF22) were exchanged into the established pET-14b-*Halo-fgf2* plasmid by double-digestion with NcoI and BamHI/NotI enzymes and ligation using T4 ligase (Fig. 1).

**Protein expression and purification of His-FGFs and Halo-FGFs**

His-FGF7, because it is toxic like native FGF7 (*Ron et al., 1993*), was transformed into BL21 (DE3) pLysS (F– ompT hsdSB(rB–, mB–) gal dcm (DE3) pLysS (CamR)) for subsequent protein expression and purification. FGF2, the other His-FGFs and Halo-FGFs were transformed into SoluBL21(F– *omp*T *hsd*SB(rB–, mB–) *gal dcm* (DE3)). The bacteria containing FGF encoding plasmids were cultured at 37 °C until the OD600 values were between 0.4 and 0.6, and then protein expression at 16 °C was induced by adding 1 mM isopropyl $\beta$-D-1-thiogalactopyranoside (IPTG). The bacteria were harvested by centrifugation at 4 °C, 14,000 g for 10 min and the pellets frozen at −80 °C.

The bacterial pellets were resuspended with the corresponding 50 mM Tris-Cl lysate buffers (pH 7.4) (Table 2), and the cells were disrupted by 5–6 cycles of sonication (30 s sonication, 60 s pause) on ice. Cell debris and insoluble proteins were removed by centrifugation at 4 °C, 30,000 g for 30 min. Then, the presence of soluble FGFs was tested by analysis of whole cells, the supernatant and pellet by separation of polypeptides on 12% (w/v) SDS-PAGE and coomassie staining.

FGF2 and His-FGF7 were purified as described before (*Xu et al., 2012*). Soluble FGF1, FGF2, FGF3, FGF10, FGF16 and FGF17, including His-FGFs and Halo-FGFs, were loaded onto a 3 mL and the other soluble FGFs were loaded onto an 8 mL column of heparin

agarose. For each FGF, different concentrations of NaCl (in 50 mM Tris-Cl pH 7.4) were used for washing and elution (Table 2) by following the previous measurements on the electrolyte sensitivity of their heparin binding assessed by Western blot (*Asada et al., 2009*). The yields of His-FGFs and Halo-FGFs were quantified by measuring the absorbance at 280 nm and the level of impurities were estimated by analysis of coomassie stained SDS-PAGE gels with ImageJ-Analyze-Gels (*Ferreira & Rasband, 2012*). The soluble His-FGFs eluted from heparin affinity chromatography were further purified by $Ni^{2+}$ affinity chromatography. Due to the negative charge on the surface of HaloTag and positive charge on the surface of FGFs, Halo-FGFs could bind to both cation- and anion-exchange stationary phases. Thus, Halo-FGF1, Halo-FGF2, Halo-FGF3, Halo-FGF7 and Halo-FGF10 were purified by chromatography on a 5 mL HiTrap Q HP column. Samples were applied in 0.15 M NaCl in PB buffer (2.7 mM KCl, 10 mM $Na_2HPO_4$, 1.8 mM $KH_2PO_4$, pH 7.4) and eluted with a gradient running to 0.8 M NaCl in the same buffer. Halo-FGF6 and Halo-FGF20 were purified by chromatography on a 3 mL column of CM Sepharose Fast Flow followed by a 3 mL column of DEAE Sepharose Fast Flow. Samples were again applied in 0.15 M NaCl in PB buffer and eluted with 0.4 M NaCl in the same buffer. The purified His-FGFs and Halo-FGFs were analysed by 12% (w/v) SDS-PAGE followed by coomassie staining.

## Purification of FGFs by removing HaloTag from Halo-FGFs

To test the accessibility of the TEV cleavage site, some Halo-FGFs, including Halo-FGF2, Halo-FGF17, Halo-FGF6, Halo-FGF8 and Halo-FGF22 eluted with high concentration of NaCl in 50 mM Tris buffer from heparin agarose chromatography and Halo-FGF20 purified with heparin, DEAE and CM chromatography, were incubated with 2.5% (mol/mol) TEV protease at 4 °C overnight. In cases where the digestion products were cloudy, they were clarified by centrifugation for 30 min at 13,000 g, 4 °C. Samples were then analysed on a 12% (w/v) SDS-PAGE. The supernatants of the TEV digestions of Halo-FGF6 and of Halo-FGF20 were applied onto a 2 mL heparin agarose column, and washed as before (Table 2). FGF6 and FGF20 were eluted with PB buffer containing 1 M NaCl or 0.1 M arginine and 1 M NaCl, respectively. After TEV digestion, FGF17 was further purified on a 1 mL HiTrap SP HP cation-exchange column by washing with 0.3 M NaCl and eluting with 1 M NaCl, both in 50 mM Tris-Cl, pH 7.4. All of the fractions from the purification steps were analysed by 12% (w/v) SDS-PAGE.

## Cell culture

Rama 27 cells were cultured in DMEM medium containing 10% (v/v) FCS, 4 mM L-glutamine, 0.75% (w/v) sodium bicarbonate, 50 ng/mL insulin and 50 ng/mL hydrocortisone (*Rudland, Twiston Davies & Tsao, 1984*). HaCaT cells were cultured in the same medium, but without insulin and hydrocortisone (*Boukamp et al., 1988*). Cell number was measured with a Z1 coulter particle counter (Beckman Coulter, High Wycombe, UK).

## Measurement of p44/42$^{MAPK}$ phosphorylation

Cells were cultured in 3 cm dishes until near confluence. Then, the dishes were washed twice with phosphate-buffered saline (PBS) and 2.5 mL step-down medium (SDM: DMEM with 250 ng BSA, 0.75% (w/v) sodium bicarbonate and 4 mM L-glutamine) was added for 24 h (Rama 27 cells) or 48 h (HaCaT cells). Rama 27 and HaCaT cells were then incubated with different FGFs for 15 min, as described in the figure legends. After the incubation, the cells were washed twice with ice-cold PBS and collected by scraping in 2X SDS-PAGE lysis buffer (4% (w/v) SDS, 20% (v/v) glycerol, 12% (v/v) Tris-Cl (pH 6.8), 2.5% (v/v) $\beta$-mercaptoethanol, 0.02% (w/v) bromophenol blue, 1 tablet of protease inhibitor and 6.8 mL distilled water). The cell lysates were heated for 10 min at 98 °C prior to SDS-PAGE.

## Western blot

After separation by 10% (w/v) SDS-PAGE, polypeptides were transferred onto a PVDF membrane. The membrane was blocked with 5% (w/v) skimmed milk in 1X TBST (50 mM Tris-Cl, 150 mM NaCl and 0.05% Tween-20 (v/v), pH 7.5) for 2 h. After two washes with TBST, the membrane was incubated with phospho-p44/42$^{MAPK}$ antibody (1:1,000 dilution in TBST) on a shaker overnight at 4 °C. Secondary anti-mouse antibody (1:1,000 dilution) was added to the membrane after three washes with TBST, 5 min each, for 1 h at room temperature. Following three washes with TBST to remove the excess secondary antibody, the membrane was covered with 1 mL ECL solution and signal was detected with Hyperfilm. The same membrane was stripped with 2.5% (w/v) SDS in TBST at 50 °C for 1 h and reblocked as above, before probing with $\beta$-actin antibody (1:10,000 dilution). The Western blot band intensity was quantified in the same way as SDS-PAGE bands and the signal intensities of phospho-p44/42$^{MAPK}$ were normalised by dividing by the intensity of the band corresponding to $\beta$-actin and then by that of the BSA control samples.

## Cell growth assay

Cell growth in Rama 27 fibroblasts was measured as before (*Smith, Winslow & Rudland, 1984*). Rama 27 cells were dispensed into a 24 well cell culture plates at 2,000 cells/well. After 24 h the cells were washed twice with PBS and cultured in SDM, as described for the p44/42$^{MAPK}$ phosphorylation assay for 24 h. The SDM was then replaced and the appropriate proteins added, as described in the figure legend. After 68 h incubation, cells were trypsinised and the number of cells counted.

## RESULTS AND DISCUSSION

### Expression of soluble FGFs

Based on their relative expression and solubility properties, the FGFs were split into three different groups: FGFs that expressed well as soluble proteins (Group 1: FGF1, FGF2 and FGF10), FGFs that expressed at a low level (Group 2: FGF3 and FGF7), and FGFs that were insoluble when expressed in *E. coli* (Group 3: FGF6, FGF8, FGF16, FGF17, FGF20 and FGF22).

## Group 1: soluble FGFs

After induction, bands corresponding to the expected molecular size of His-FGF1, FGF2 and His-FGF10 were apparent in the whole cell lysates (Figs. 2A, 2C and 2E, lane L, green arrow). His-FGF1 and His-FGF10 were expressed at a higher level than FGF2 in *E. coli* SoluBL21. After centrifugation of the cell lysates, bands corresponding to the molecular size of all three FGFs were mainly recovered in the soluble fraction (supernatant), rather than in the insoluble fraction (pellet; Figs. 2A, 2C and 2E, lanes S and P). Chromatography of the supernatants on heparin demonstrated that little of the expressed protein was present in the flow-through fraction (Figs. 2A, 2C and 2E, lane T). Weak bands corresponding to His-FGF1 and His-FGF10, but not FGF2, were observed in the wash fraction (Figs 2A and 2E, lane Wa), which may represent aggregated or less well-folded protein. The majority of the three FGFs was recovered in the high NaCl eluate (Figs. 2A, 2C and 2E, lane Hep), which demonstrated that these soluble FGFs bound heparin strongly. This indicated that they were likely to be properly folded, because the canonical, highest affinity heparin binding site of FGFs depends on the tertiary structure of the proteins (*Xu et al., 2012*).

The bands corresponding to Halo-FGF1, Halo-FGF2 and Halo-FGF10 were clearly observed in the whole cell lysates and these proteins were all highly expressed in SoluBL21 cells (Figs. 2B, 2D and 2F, lane L, red arrow). Similarly to the His-FGF1, FGF2 and His-FGF10, after centrifugation of the whole cell lysates, the bands corresponding to the three Halo-FGFs were observed in the soluble fractions (Figs. 2B, 2D and 2F, lanes S and P). Chromatography of the soluble fractions on heparin indicated that most of Halo-FGF2 and Halo-FGF10 had bound to the column, but there was a substantial amount of Halo-FGF1 in the flow-through (Figs. 2B, 2D and 2F, lane T). This may be due to the capacity of the column for Halo-FGF1 being lower than for His-FGF1. All three Halo-FGFs were eluted from the heparin affinity column at the expected NaCl concentration (Figs. 2B, 2D and 2F, lane Hep).

The yield of Halo-FGF1 and Halo-FGF10 was similar to that of the corresponding his-tagged proteins (Table 3). However, since the Halo-FGF proteins are considerably larger than the corresponding His-tagged FGF1 and FGF10, this represents a decrease in the molar amounts of FGF produced. In contrast, the yield of Halo-FGF2 was 4-fold higher (Table 3), which is only partly accounted for by the increased size of the fusion protein. The low yield of full-length FGF2 has been ascribed to the presence of secondary structure at the 5′ end of the FGF2 mRNA (*Knoerzer et al., 1989*), and the presence of the upstream HaloTag sequence may mitigate this effect.

## Group 2: low expression proteins

The expression of His-FGF3 was weak, as was that of His-FGF7 (expressed in BL21 DE3 pLysS) due to its toxicity (*Ron et al., 1993*) (Figs. 3A and 3C, lane L, S and P, green arrow). Heparin chromatography of the supernatants demonstrated that the yields of soluble His-FGF3 and His-FGF7 were quite low (Figs. 3A and 3C, lane Hep; Table 3).

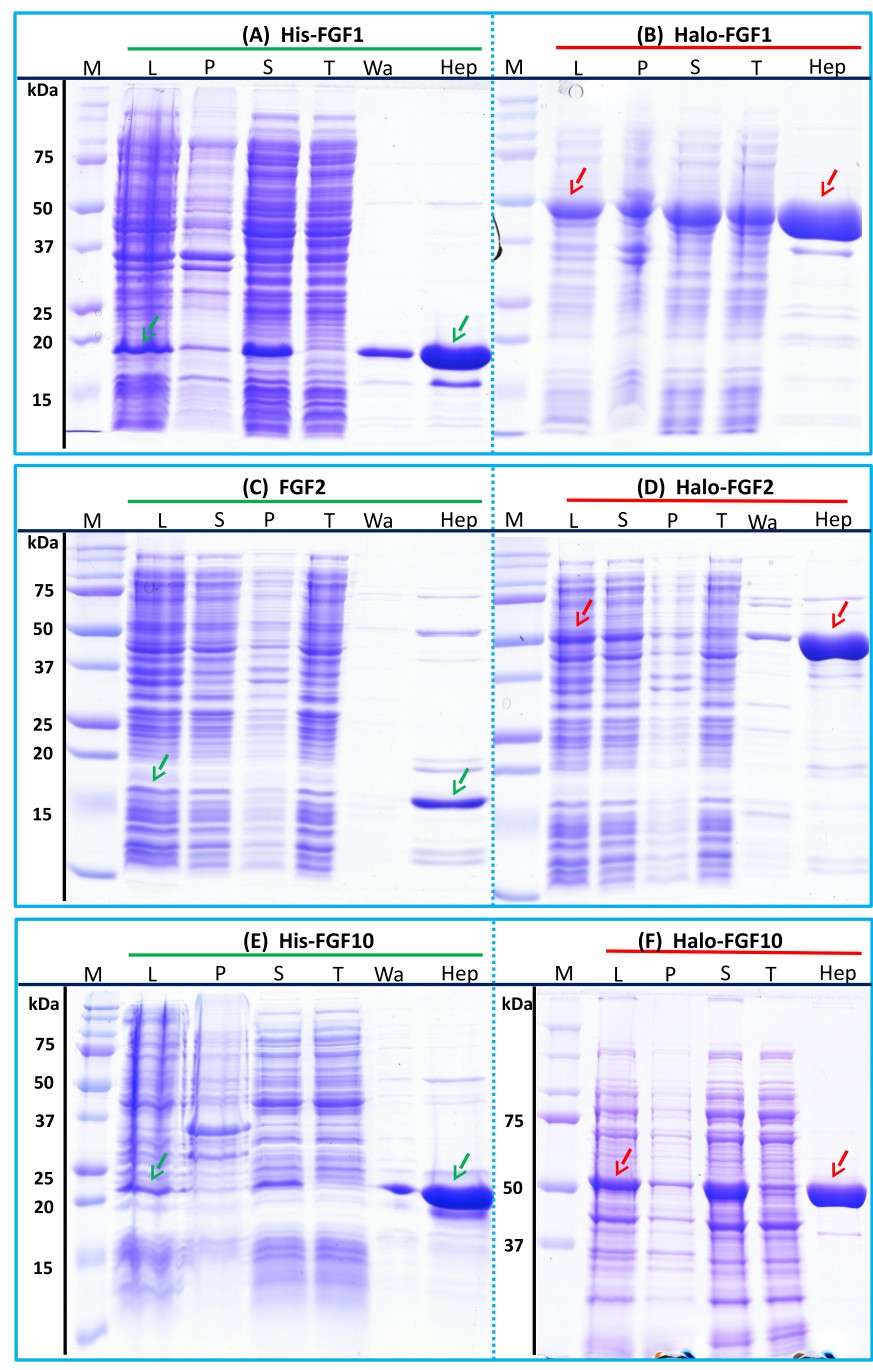

**Figure 2 Expression and heparin affinity purification of His-FGF1, FGF2, His-FGF10, Halo-FGF1, Halo-FGF2 and Halo-FGF10.** Following induction of expression with IPTG, cells were lysed by sonication and the insoluble material collected by centrifugation. The supernatant was subjected to heparin-affinity chromatography and samples were then analysed by SDS-PAGE and coomassie staining. Lane M, markers; L, sonicated whole cell lysate; P, pellet following centrifugation of lysate; S, corresponding supernatant; T, unbound, flow-through fraction from heparin-affinity chromatography; Wa, wash of heparin-affinity column (Table 2); Hep, high NaCl eluate of heparin-affinity column (Table 2). Green arrows: FGF or His-FGF; red arrows: Halo-FGF.

**Table 3 Summary of the molecular sizes and yields of His-FGFs and Halo-FGFs.** The molecular weight of the proteins was calculated from their amino acid sequence. The concentrations and volumes of His-FGFs and Halo-FGFs recovered from heparin affinity chromatography were measured. The impurities identified by SDS-PAGE were quantified using ImageJ relative to the band corresponding to His-FGF and to Halo-FGF and the amount of protein in the eluate from heparin chromatography adjusted accordingly, to provide an estimate of the yield.

| FGFs | Molecular weight (kDa) | | Yield (mg/L) | |
|---|---|---|---|---|
| | HisTag | HaloTag | HisTag | HaloTag |
| FGF1 | 19.1 | 50.9 | 14 | 16 |
| FGF2 | 17.3 | 52.2 | 2.5 | 11 |
| | No Tag | | No Tag | |
| FGF3 | 28.2 | 60.0 | 0.5 | 11 |
| FGF6 | 22.3 | 54.1 | n.d.[a] | 27 |
| FGF7 | 22.2 | 54.0 | 0.6 | 5.6 |
| FGF8 | 25.7 | 57.5 | n.d.[a] | 1.7 |
| FGF10 | 22.7 | 54.5 | 7.7 | 9.3 |
| FGF16 | 26.9 | 58.7 | n.d.[a] | 1.0 |
| FGF17 | 25.8 | 57.6 | n.d.[a] | 1.5 |
| FGF20 | 26.9 | 58.6 | n.d.[a] | 10 |
| FGF22 | 20.5 | 52.3 | n.d.[a] | 2.0 |

**Notes.**
[a] Not detected. Insufficient soluble protein for reliable quantification.

Transformation of SoluBL21with the plasmid encoding Halo-FGF7 yielded the expected number of colonies, indicating that the fusion protein was not toxic. Bands corresponding to the molecular size of Halo-FGF3 and Halo-FGF7 were observed in the cell lysates (Figs. 3B and 3D, lane L, red arrow) and in the soluble fraction obtained after centrifugation, whereas the pellet has relatively weaker bands (Figs. 3B and 3D, lanes P and S), indicating that Halo-FGF3 and Halo-FGF7 were soluble. Heparin chromatography of the soluble factions demonstrated that large amounts of Halo-FGF3 and Halo-FGF7 retained their heparin binding interaction with the polysaccharide (Figs. 3B and 3D, lane Hep).

The yields of Halo-FGF3 and of Halo-FGF7 were 21-fold and 9-fold greater than of the corresponding His-tagged FGF (Table 3). Thus, the presence of the HaloTag N-terminal fusion increased the amounts of FGF3 and FGF7 substantially, even after taking into account the larger size of these fusion proteins (Table 3).

## Group 3: insoluble proteins

His-FGF6, His-FGF8, His-FGF22, His-FGF17, His-FGF16 and His-FGF20 were all expressed, albeit at different levels. After centrifugation, bands corresponding to the molecular sizes of these proteins were detected in the pellet (Fig. 4, compare lanes P and S, green arrow). Although small amounts of protein, such as bands corresponding to His-FGF6, His-FGF16 and His-FGF20, were observed in the supernatants (Fig. 4, lanes S), no protein were detected in the eluate from heparin chromatography, which might suggest these proteins were either small soluble aggregates or not properly folded. It has

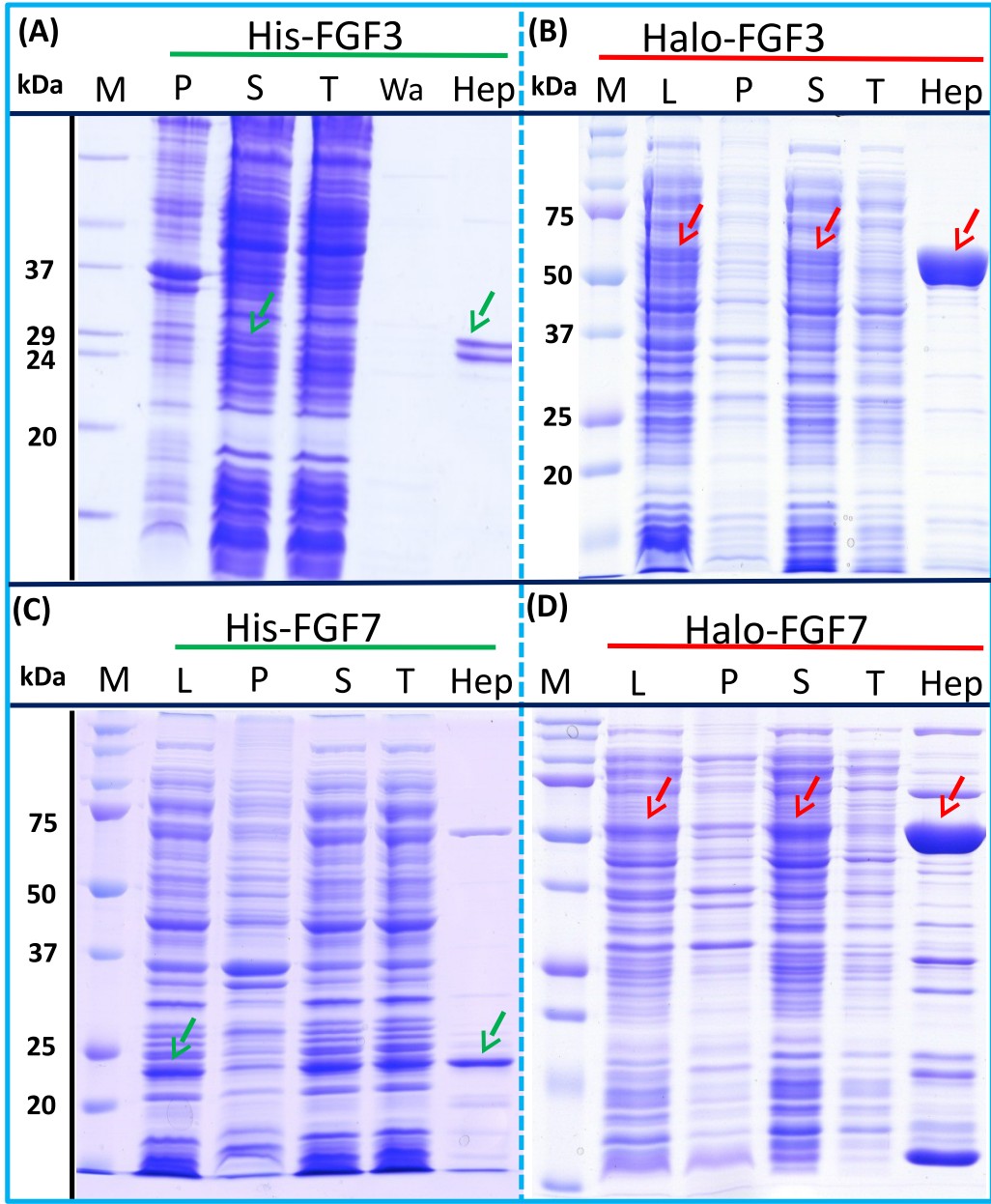

**Figure 3 Expression and heparin binding-affinity chromatography of His-FGF3, His-FGF7, Halo-FGF3 and Halo-FGF7.** Following induction of expression with IPTG, cells were lysed by sonication and the insoluble material collected by centrifugation. The supernatant was subjected to heparin-affinity chromatography and samples were then analysed by SDS-PAGE and coomassie staining. Lane M, markers; L, sonicated whole cell lysate; P, pellet following centrifugation of lysate; S, corresponding supernatant; T, unbound, flow-through fraction from heparin-affinity chromatography; Hep, high [NaCl] eluate of heparin-affinity column (Table 2). Green arrows: His-FGF; red arrows: Halo-FGF.

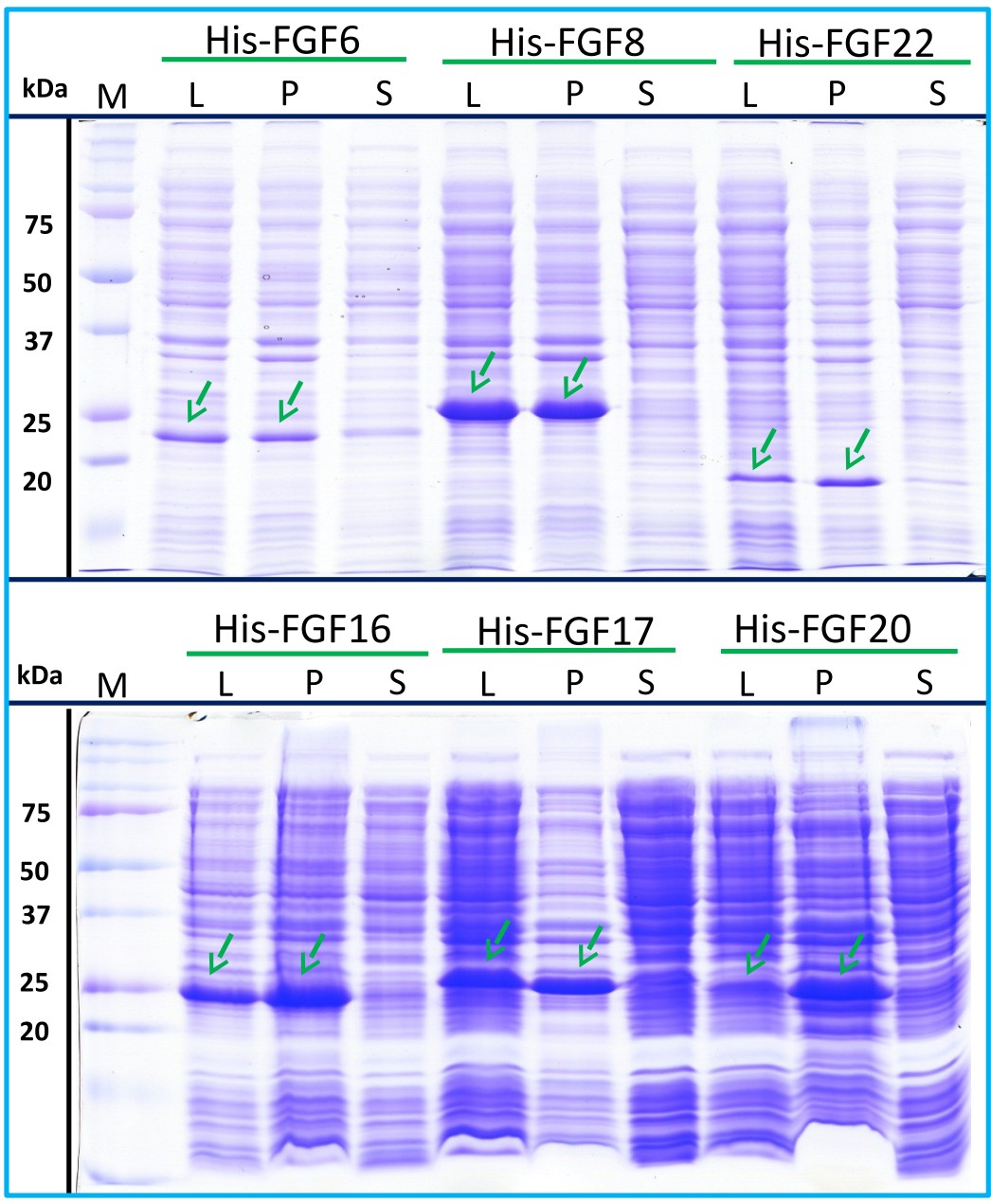

**Figure 4 Expression test of His-FGF6, His-FGF8, His-FGF22, His-FGF17, His-FGF16 and His-FGF20.** Following induction of expression with IPTG, cells were lysed by sonication and the insoluble material collected by centrifugation. The whole cell lysate, supernatant and pellet were analysed by SDS-PAGE and coomassie staining. Lane M, markers; L, sonicated whole cell lysate; P, pellet following centrifugation of lysate; S, corresponding supernatant. Green arrows: His-FGF.

been reported that FGF20 could also be solubilised by high concentrations of arginine (*Maity, Karkaria & Davagnino, 2009*), which suggests that FGF20 in the lysis buffer has a tendency to aggregate. However, arginine would compete for binding of FGFs to heparin, which reduces the utility of this approach to solubilisation.

As illustrated by SDS-PAGE, all of the bands corresponding to the molecular size of Halo-FGF6, Halo-FGF8, Halo-FGF22, Halo-FGF17, Halo-FGF16 and Halo-FGF20 were clearly observed in the whole lysates, which suggested that all six proteins expressed well in *E. coli* (Fig. 5, lanes L, red arrow), particularly Halo-FGF6, Halo-FGF17, Halo-FGF16 and Halo-FGF20. Although some material corresponding to the expected molecular size of these Halo-FGFs was observed in the pellet after centrifugation of the cell lysates (Fig. 5, lanes P), there were strong bands corresponding to Halo-FGF6, Halo-FGF16 and Halo-FGF20 and weak bands corresponding to Halo-FGF8, Halo-FGF17 and Halo-FGF22 present in the soluble fractions (Fig. 5, lanes S). Following application to a heparin affinity column, most of Halo-FGF6 in the supernatant bound to heparin and was eluted by 1 M NaCl in Tris-Cl (Fig. 5A, lanes S, T and Hep). Halo-FGF8 Halo-FGF17 and Halo-FGF22 also bound to the heparin-affinity column reasonably efficiently, whereas a considerable amount of Halo-FGF16 and Halo-FGF20 did not bind (Figs. 5B–5E, lanes S and T). All four proteins could be recovered from heparin chromatography with high concentration NaCl-containing elution buffers (Table 2) (Figs. 5B–5E, lane Hep). When the Halo-FGF20 in the flow-through fraction (Fig. 5F, lane T) was applied to a second heparin-affinity chromatography column, a large amount of Halo-FGF20 was found to bind and could be eluted (Fig. 5F, lane Hep2). A considerable amount of Halo-FGF16 also failed to bind to the heparin affinity column (Fig. 5E, lane T), though the bound protein was eluted with high NaCl (Fig. 5E, lane Hep). This suggests that the capacity of the heparin affinity column for Halo-FGF20 was exceeded. The same explanation may underlie the presence of Halo-FGF16 in the flow-through fraction, though this protein was present at a slightly lower level. However, nothing is known about the preference of either FGF16 or FGF20 for binding structures in the polysaccharide, if so these were relatively rare in heparin, the column capacity might easily be exceeded. Alternatively, the Halo-FGF16 in the flow through fraction may represent protein that is in small aggregates and/or not properly folded.

Given that the amounts of soluble His-tagged FGF6, FGF8, FGF22, FGF17, FGF16 and FGF20 were not readily detectable, it is clear that the N-terminal HaloTag fusion significantly improved the expression of soluble protein. The yield of Halo-FGF6 and Halo-FGF20 was substantial (27 mg/L and 10 mg/L, respectively, Table 3). Although a lower yield of Halo-FGF8, Halo-FGF16, Halo-FGF17 and Halo-FGF22 (1 mg/L to 2 mg/L, Table 3) was obtained, it is nevertheless sufficient for many applications, including microscopy. However, the heparin affinity purification step did not produce entirely pure protein, as judged by coomassie staining (Figs. 2, 3 and 5).

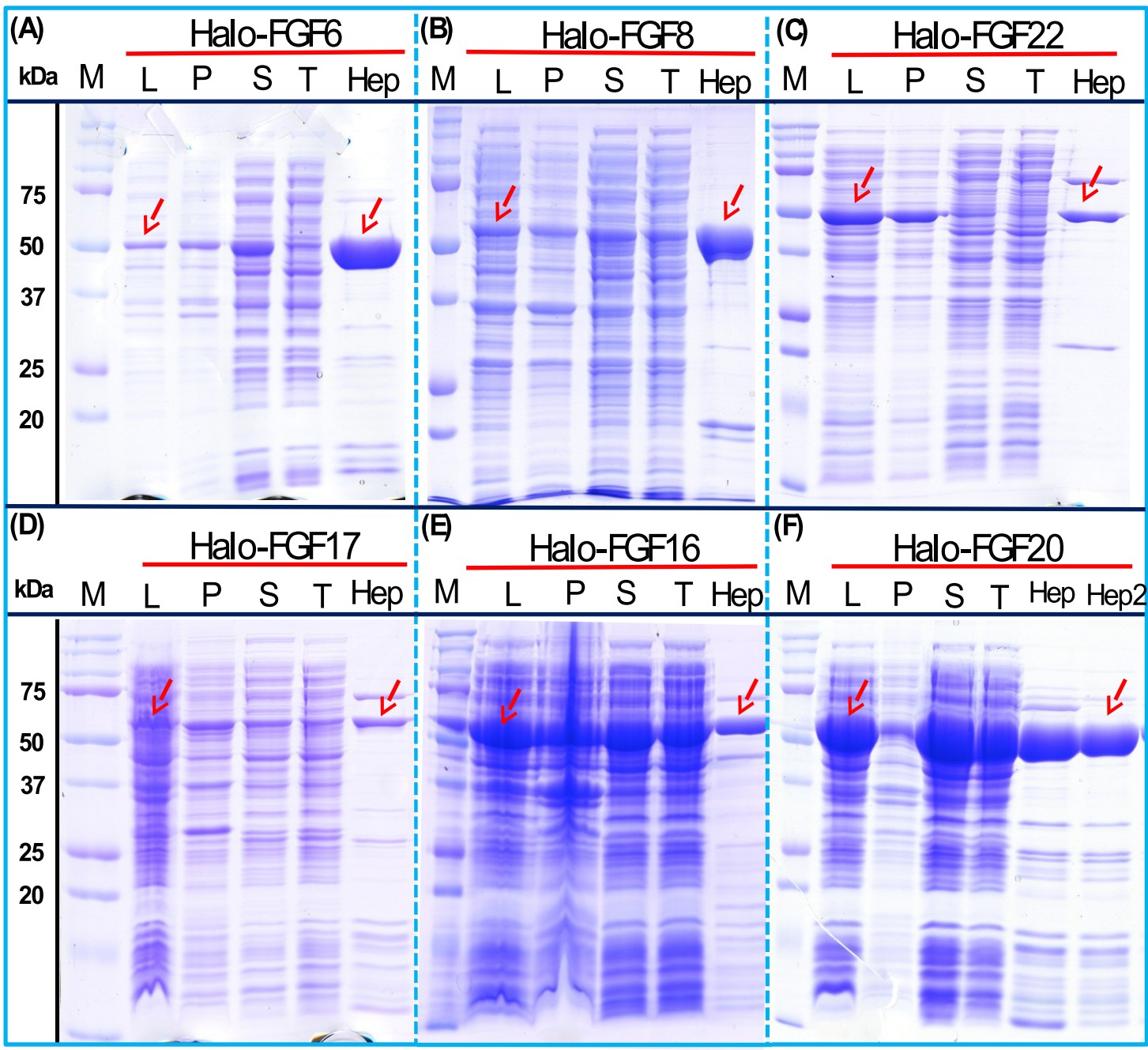

**Figure 5 Expression and heparin affinity purification of Halo-FGF6, Halo-FGF8, Halo-FGF22, Halo-FGF17, Halo-FGF16 and Halo-FGF20.** Following induction of expression with IPTG, cells were lysed by sonication and the insoluble material collected by centrifugation. The supernatant was subjected to heparin-afinity chromatography and samples were then analysed by SDS-PAGE and coomassie staining. Lane M, markers; L, sonicated whole cell lysate; P, pellet following centrifugation of lysate; S, corresponding supernatant; T, unbound, flow-through fraction from heparin-affinity chromatography; Hep, high NaCl eluate of heparin-affinity column (Table 2). Red arrows: Halo-FGF.

## Further purification of some Halo-FGFs

Four Halo-FGFs, Halo-FGF1, Halo-FGF7, Halo-FGF6 and Halo-FGF20 were chosen to determine whether the Halo-FGFs could be easily subjected to further purification, since there was clear evidence for impurities following heparin-affinity chromatography. Halo-FGF1 and Halo-FGF7 were successfully purified by Q anion-exchange chromatography (Figs. 6A and 6B, lane Q), which depends on the acidic isoelectric point of the HaloTag (pI: 4.77). For Halo-FGF6 and Halo-FGF20, advantage was taken of the acidic HaloTag and positive surfaces of FGFs, to enable a two-step ion-exchange purification of the eluate from heparin-affinity chromatography, using both DEAE anion and CM cation ion-exchange chromatography (Figs. 6C and 6D, lane DEAE and CM). The isolated Halo-FGFs are relatively pure, as is shown on the gels (Fig. 6).

## Purification of FGFs by removing the HaloTag with TEV protease

The inclusion of a TEV site between the sequence of the HaloTag and FGF proteins provides a means to remove the HaloTag fusion partner in those instances where the HaloTag is not required for analysis (or when it may interfere with such analyses). Halo-FGF2 was first incubated with TEV protease to test whether the fusion protein could be cleaved by TEV. SDS-PAGE of the TEV digestion product of Halo-FGF2 shows that almost all of the protein was cleaved into the 35 kDa HaloTag (Fig. 7A, red arrow) and the 18 kDa FGF2 (Fig. 7A, green arrow). Thus, the cleavage site is fully accessible to TEV protease. Both Halo-FGF17 and Halo-FGF20 were also well digested by TEV protease and subsequently soluble FGF17 (Fig. 7B, green arrow) and FGF20 (Fig. 7C, green arrow) were purified by cation-exchange and heparin chromatography, respectively.

Most of FGF6 (Fig. 7D, lane $W_{Dig}$, green arrow) and FGF22 (Fig. 7F, lane $W_{Dig}$, green arrow) and a small proportion of FGF8 were also released from HaloTag (Figs. 7D–7F, lane $W_{Dig}$ and S, red arrow), but these proteins were observed to aggregate upon cleavage. This suggested that these proteins were not very stable, at least in the buffer conditions used here, and required the HaloTag N-terminal fusion to remain soluble. The soluble FGF6 released by cleavage (Fig. 7D, lane S, green arrow) was applied to a heparin affinity column, but was observed to be concentrated at the top of the column where it formed a white aggregate. Very little protein was eluted with 1 M NaCl in PB buffer (Fig. 7D, lane E, green arrow). The disappearance of FGF8 and FGF22 in the soluble fractions after TEV digestion (Figs. 7E and 7F, lane S) showed that these two proteins were also not very soluble in the present buffer conditions without the HaloTag fusion partner.

## Biological activities of FGFs and Halo-FGFs on Rama 27 fibroblasts and HaCaT keratinocytes

FGF1, FGF2 and FGF6 have a preference for FGFR1c (*Zhang et al., 2006a*), the predominant receptor expressed by Rama 27 fibroblasts (*Delehedde et al., 2000*; *Zhu et al., 2010*). When Rama 27 cells were stimulated with 25 pM FGF2 for 15 min, strong bands corresponding to dually phosphorylated p44/42$^{MAPK}$ were apparent (Fig. 8A), as observed previously (*Delehedde et al., 2000*; *Zhu et al., 2010*). Halo-FGF2 caused a similar

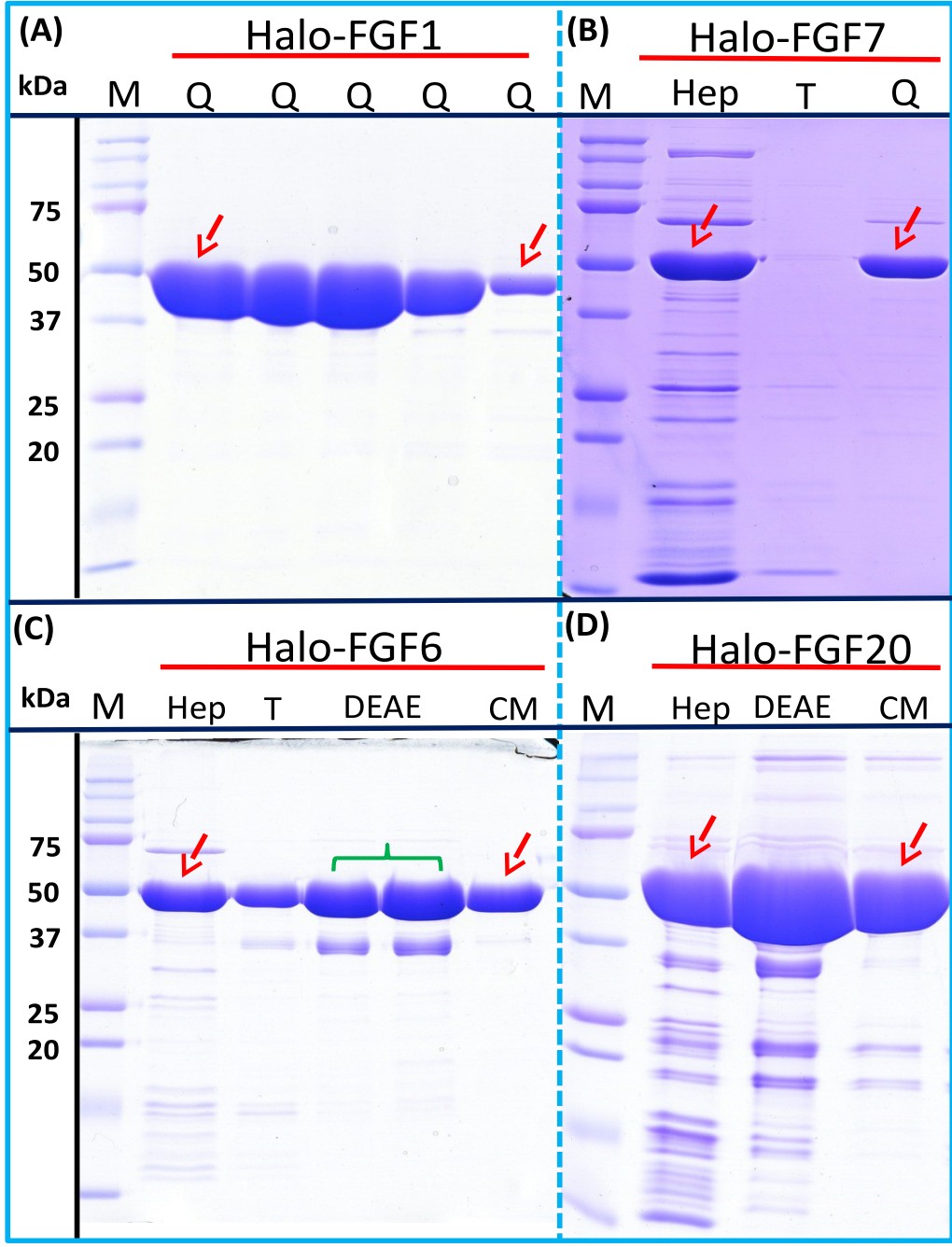

**Figure 6 Further purification of the heparin affinity eluate of Halo-FGF1, Halo-FGF6, Halo-FGF7 and Halo-FGF20 by ion-exchange chromatography.** The soluble Halo-FGF1 and Halo-FGF7 eluted from heparin chromatography was purified using Q ion-exchange chromatography, while CM and DEAE ion-exchange chromatography were used to purify Halo-FGF6 and Halo-FGF20. Lane M, markers; Hep, eluate from heparin chromatography, Figs. 2A, 3D, 5A and 5F; T, unbound, flow-through fraction from ion-exchange chromatography; Q, peak fractions collected from Q HP chromatography; DEAE eluate from DEAE chromatography, two identical samples; CM, eluate from CM chromatography. Red arrows: Halo-FGF.

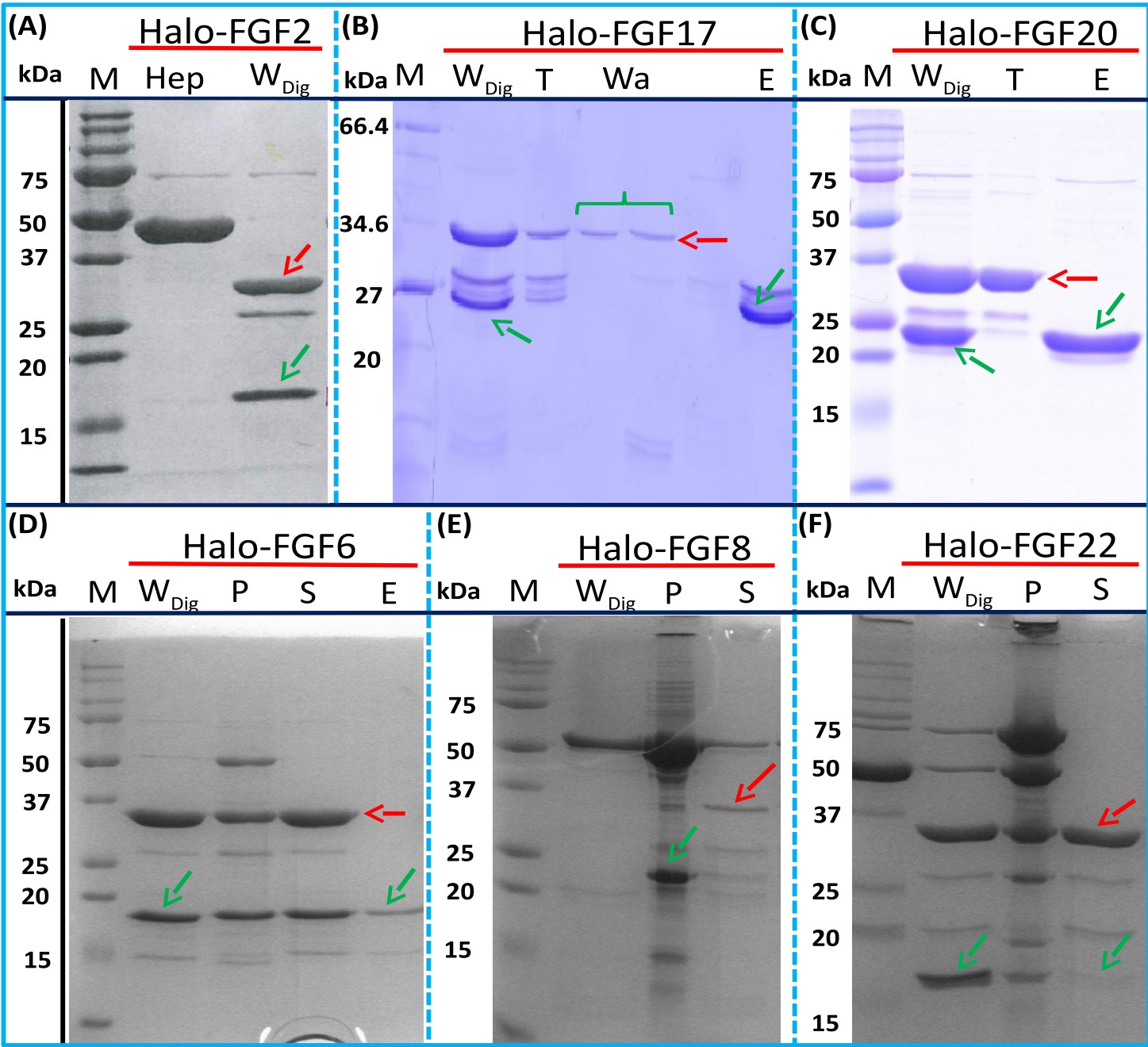

**Figure 7 Cleavage of Halo-FGFs by TEV and purification.** The eluates of Halo-FGF2, Halo-FGF17, Halo-FGF6, Halo-FGF8 and Halo-FGF22 from heparin-affinity chromatography and the Halo-FGF20 purified by heparin and ion-exchange chromatography were digested by TEV protease to separate the HaloTag and the FGF. Halo-FGF6, Halo-FGF8 and Halo-FGF22 became turbid after digestion and these samples were clarified by centrifugation. Then, the samples containing FGF6 and FGF20 were subjected to heparin chromatography and that of FGF17 to SP HP cation-exchange chromatography. Lanes M, markers; Hep, eluate from heparin chromatography; $W_{Dig}$, whole digestion product of Halo-FGFs purified by heparin chromatography; T, unbound, flow-through fraction from heparin chromatography; Wa, wash of SP HP cation-exchange chromatography; P, pellet following centrifugation of product of TEV digestion; S, supernatant after the centrifugation; E, high NaCl eluate of heparin or SP cation-exchange chromatography. Green arrows: FGF; red arrows: HaloTag.

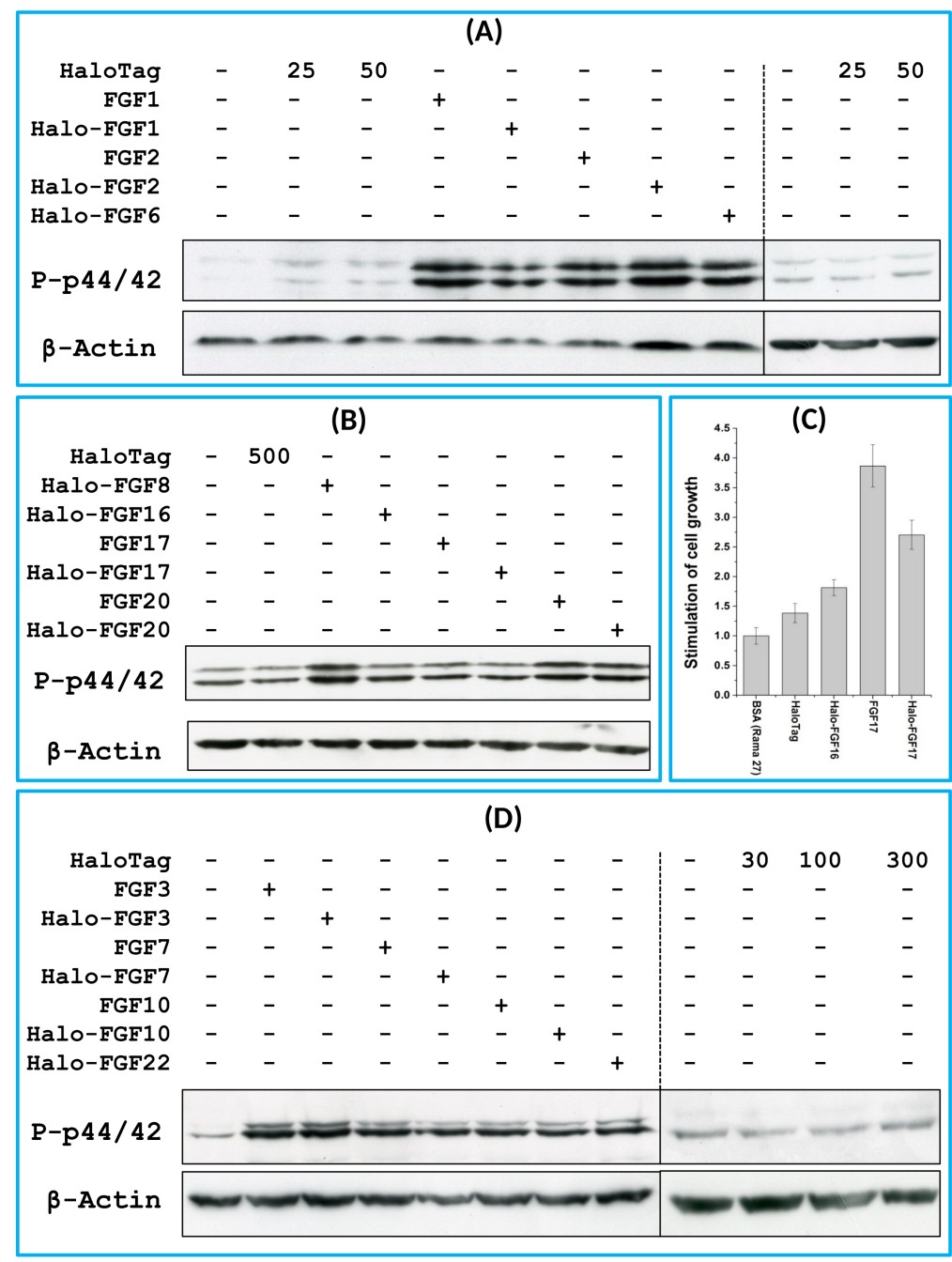

**Figure 8 Activities of FGFs on Rama 27 fibroblasts and HaCaT keratinocytes.** Cells were grown in 3 cm diameter dishes (Western blots) or 24-well plates growth assay, as described in the Materials and Methods. After incubation in SDM for 24 h (Rama 27) or 48 h (HaCaT), cells were stimulated with the FGF protein for 15 min (Western blot) or 68 h (cell growth assay). (A) Stimulation of p44/42$^{MAPK}$ phosphorylation by 25 pM HaloTag, FGF2, Halo-FGF2 and 50 pM HaloTag, FGF1, Halo-FGF1 and Halo-FGF6 in Rama 27 fibroblasts. (B) Stimulation of p44/42$^{MAPK}$ phosphorylation by 500 pM HaloTag, Halo-FGF8, Halo-FGF16, FGF17, Halo-FGF17, FGF20 and Halo-FGF20 in Rama 27 fibroblasts. (C) Stimulation of cell growth of Rama 27 fibroblasts by 10 nM Halo-FGF16, FGF17 and Halo-FGF17. (D) Stimulation of p44/42$^{MAPK}$ phosphorylation by 300 pM His-FGF3, Halo-FGF3 and Halo-FGF22, 30 pM His-FGF7, Halo-FGF7, His-FGF10 and Halo-FGF10 in HaCaT cells.

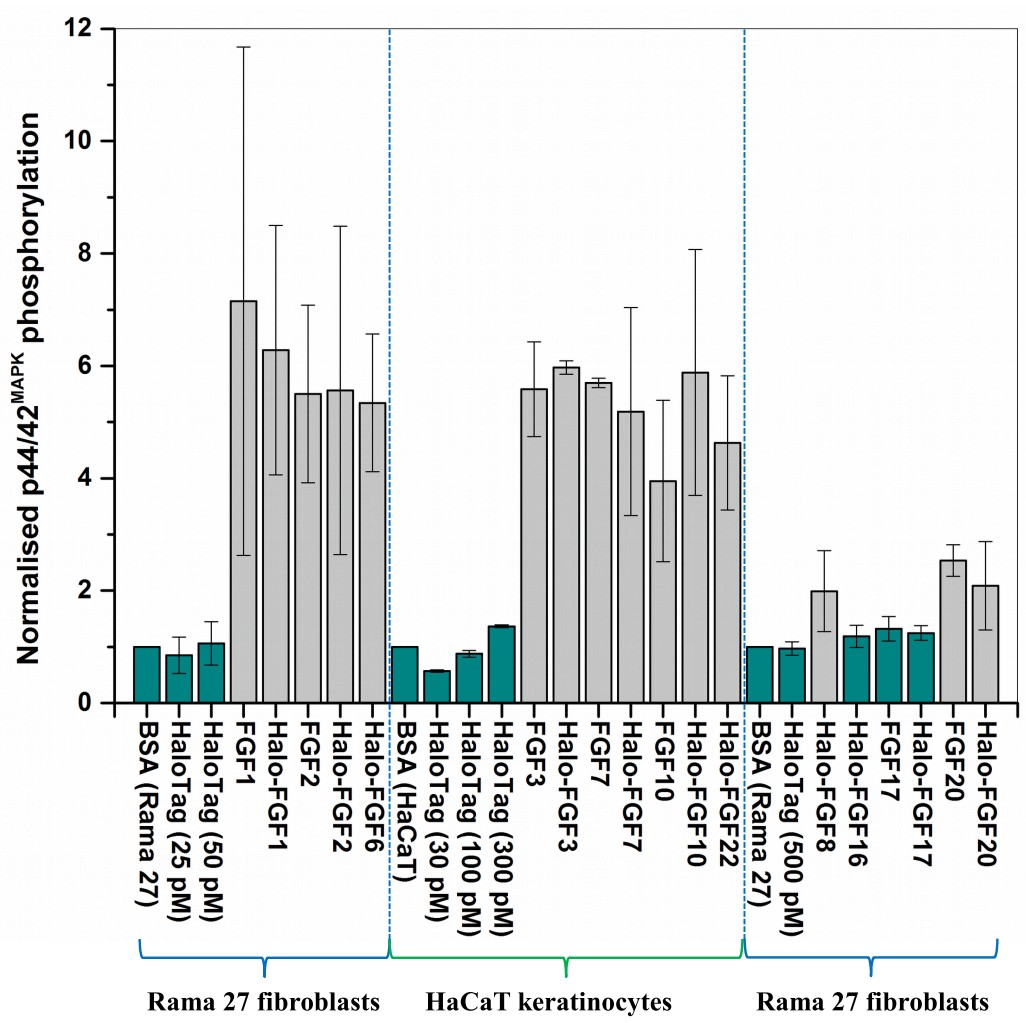

**Figure 9 Quantification of p44/42$^{MAPK}$ phosphorylation.** The band intensities from two experiments were quantified with imageJ and normalised to the BSA control to compare the similarities and differences of stimulation of phosphorylation p44/42$^{MAPK}$ by different FGFs. Results are the mean with the actual values from two independent experiments.

stimulation of phosphorylation of p44/42$^{MAPK}$ (Fig. 8A). In contrast, the 25 pM or 50 pM HaloTag protein alone did not appreciably stimulate p44/42$^{MAPK}$ phosphorylation. Therefore, the activity of Halo-FGF2 in this assay is equivalent to that of FGF2 (Fig. 9). In the case of FGF1, the N-terminal HaloTag also did not affect the ability of the growth factor to stimulate the phosphorylation of p44/42$^{MAPK}$ (Fig. 8A). FGF6 is not soluble without the HaloTag, so only the activity of the fusion protein could be tested, and it was found to stimulate the phosphorylation of p44/42$^{MAPK}$ to an extent similar to that observed with FGF1 and FGF2 (Fig. 8A). Since FGF6 has the same receptor preference as FGF1 and FGF2 (*Zhang et al., 2006a*), this suggests Halo-FGF6 was fully active.

FGF8, FGF16, FGF17 and FGF20 have a preference for FGFR3c, but they are also able to activate FGFR1c, though higher concentrations of growth factor are required to elicit

activity (*Zhang et al., 2006a*). When 500 pM HaloTag was added to the cells, there was no detectable increase in phosphorylation of p44/42$^{MAPK}$, whereas Halo-FGF8, Halo-FGF20 and FGF20 at concentrations comparable to those used in previous work (*Zhang et al., 2006a*) were all found to stimulate the phosphorylation of p44/42$^{MAPK}$ (Fig. 8B). In contrast, Halo-FGF16, Halo-FGF17 and FGF17 did not cause a detectable increase in phosphorylation of p44/42$^{MAPK}$ (Fig. 9). These data indicate that Halo-FGF8, FGF20 and Halo-FGF20 have biological activities on Rama 27 fibroblasts. The absence of stimulation of phosphorylation of p44/42$^{MAPK}$ by Halo-FGF16 may reflect the fact that the ability of this FGF to activate FGFR1c is considerably lower than that of FGF8, FGF17 and FGF20 (*Zhang et al., 2006a*). However, the absence of stimulation of phosphorylation of p44/42$^{MAPK}$ by FGF17 and Halo-FGF17 is more puzzling. One explanation may be that FGF16, and perhaps FGF17, do not cause the FGFR to activate strongly early biochemical signals that converge on p44/42$^{MAPK}$. To test this, the capacity of Halo-FGF16, Halo-FGF17 and FGF17 to stimulate cell growth was measured in Rama 27 fibroblasts. The results show that 10 nM HaloTag only weakly stimulated the growth of Rama27 fibroblasts. Halo-FGF16 caused the number of cells to double compared to the negative control, and this level was significantly ($p = 0.015$, Tukey test, OriginPro 9) above that observed in the presence of HaloTag alone (Fig. 8C). Halo-FGF17 and FGF17 were even more effective, as they caused a 3- to 4-fold increase in the number of cells (Fig. 8C). These results demonstrated that Halo-FGF16, FGF17 and Halo-FGF17 possess biological activities of similar potency as observed by others in growth assays (*Zhang et al., 2006a*).

The activity of members of the FGF7 subfamily were tested on HaCaT keratinocytes, as this cell type expresses the cognate receptor for these FGFs, FGFR2b (*Ron et al., 1993*). HaCaT cells have previously been shown to express more p42$^{MAPk}$ than p44$^{MAPk}$ (*Delehedde et al., 2002*). The data show clearly that HaloTag alone did not stimulate the phosphorylation of p44/42$^{MAPK}$ (Fig. 8D). In contrast, FGF3, FGF7 and FGF10, and the corresponding HaloTag fusion proteins stimulated p44/42$^{MAPK}$ phosphorylation (Fig. 8C). FGF22, which is only soluble as a HaloTag fusion protein, also stimulated p44/42$^{MAPK}$ phosphorylation to an extent similar to that seen with the other members of the subfamily (Fig. 8D). Thus, these Halo-FGFs retain full biological activity in this assay.

## CONCLUSION

In this study, we identified four useful properties of N-terminal HaloTag fusions for the production of biologically active FGFs: (i) using the HaloTag can increase the yield of low expression FGFs; (ii) the HaloTag rendered FGF7 non-toxic; (iii) for the insoluble FGFs, the HaloTag enabled *E.coli* to express more soluble protein at low induction temperatures and maintain solubility during isolation and storage; (iv) a consequence of the low isoelectric point of HaloTag was that anion-exchange chromatography could be used as an orthogonal step in the purification of the Halo-FGFs. However, there are clearly limitations; for example, some of the FGFs did not retain solubility following cleavage from the HaloTag. This may reflect the fact that no single solubilisation tag is a universal panacea for resolving the problems of protein expression (*Costa et al., 2014*). Nevertheless, because

the HaloTag can enhance expression of soluble protein and provide a means to label FGF protein with different fluorescent dyes and quantum dots, e.g., *Los et al. (2008)*; *Zhang et al. (2006b)* it is clearly a versatile and useful tool for these two purposes and, therefore, worthwhile exploring as a part of experimental strategy with these aims.

## ACKNOWLEDGEMENT

Xianqing Mao would like to thank Monika Dieterle for help with cloning Halo-FGF3.

### Funding

The Cancer and Polio Research Fund and North West Cancer Research provided financial support. The funders had no role in study design, data collection and analysis, decision to publish, or preparation of the manuscript.

### Grant Disclosures

The following grant information was disclosed by the authors:
Cancer and Polio Research Fund and North West Cancer Research.

### Competing Interests

The authors declare there are no competing interests.

### Author Contributions

- Changye Sun and Yong Li conceived and designed the experiments, performed the experiments, analyzed the data, contributed reagents/materials/analysis tools, wrote the paper, prepared figures and/or tables, reviewed drafts of the paper.
- Sarah E. Taylor and Xianqing Mao performed the experiments, contributed reagents/materials/analysis tools, reviewed drafts of the paper.
- Mark C. Wilkinson performed the experiments, reviewed drafts of the paper.
- David G. Fernig conceived and designed the experiments, analyzed the data, wrote the paper, reviewed drafts of the paper.

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
