# Peer review of "HaloTag is an effective expression and solubilisation fusion partner for a range of fibroblast growth factors"

_PeerJ, doi:10.7717/peerj.1060_

## Round 0.1 · original submission · Major Revisions

Please be aware that we would like to see data that shows the activity of the recombinant protein in cell based or receptor assays. There is no evidence provided in the current version of the manuscript to state that these proteins are as active as FGF produced by other methodologies. As a consequence, you will see that one reviewer did not find your work a major advancement.

Reviewer 1 ·

Basic reporting

There is nothing novel in this manuscript. The authors have expressed several FGFs using Halo tag and authors conclude that Halo tag provides a tool for expression of soluble proteins

Experimental design

The authors have expressed FGFs using the tag. Again there is nothing novel in this that deserves publication

Validity of the findings

No comments

Additional comments

In general, the manuscript describes expression of FSFs using Halo which increases expression of the soluble protein. It is difficult to understand the novelty of this entire exercise or the importance of the work.

Reviewer 2 ·

Basic reporting

The expression of recombinant proteins in E. coli can be a challenging task in general and especially difficult for proteins of the FGF family due to solubility problems. The authors of this manuscript aimed to apply the recently invented HaloTag technology to covalently label FGFs of interest. Obviously, improved solubilty and expression as well as labeling could provide better means to further analyze e.g. ligand binding and subsequent ligand/receptor complex internalization.

Experimental design

Comments/Questions:

1. It would be interesting to know whether the purified and tagged FGFs retain biological activity. For this purpose, the authors could demonstrate e.g. for FGF2 downstream signaling or neurite outgrowth (CG8 assay) in a suitable bioassay.

2. What was the strategy for selecting a number of FGFs but not others? For example, did the authors try to express FGF23 or FGF4?

Validity of the findings

3. With regard to FGF-2, have the authors tried to tag and purify high-molecular weight isoforms?

Minor points:

4. The authors should reference or provide the genotypes of E. coli pLysS and SoluBL21 and other bacterial strains used.

Reviewer 3 ·

Basic reporting

No Commends

Experimental design

No Comments

Validity of the findings

No Comments

Additional comments

Authors showed that N-terminal Halo Tag fusion had a substantial effect on the expression of the more recalcitrant FGFs.

Main point, how do we know that the fusion Halo Tag-FGFs were properly folded particularly that Halo Tag has low isoelectric point. Moreover some of these FGFs did not retain solubility following cleavage from the Tag.
It seems that not much is known about activities of purified FGFs as a fusion protein and after digestion. Using affinity chromatography and linear gradient they could check out differences in binding to heparin column. And using cells expressing FGFRs find out if the purified growth factors bind (and activated) high affinity FGF receptors.

---

## Round 0.2 · accepted · Accept

The paper has been re-reviewed and found to be suitable for acceptance. The response letter also states clearly your views on some of the comments made in the earlier review.

Reviewer 2 ·

Basic reporting

no comments

Experimental design

no comments

Validity of the findings

no comments

Additional comments

The criticisms have been addressed very well by including new experimental data and other improvements of the manuscript. I recommend publication.